# Clinical impact of BioFire Blood Culture Identification 2 implementation in patients with *Staphylococcus* bacteremia

Colin H. Duell,[1] Tyler Ackley,[1] Kristin E. Linder,[1] Joseph L. Kuti,[2] Alivia Castle,[3] Rosanna Li[4]

**ABSTRACT**   The BioFire Blood Culture Identification 2 (BCID2) PCR panel allows for rapid identification of blood pathogens, including Staphylococci other than *Staphylococcus aureus* (SOSA) and methicillin-susceptible *S. aureus* (MSSA). The objective of this study was to evaluate how BCID2 implementation impacted duration of vancomycin therapy, clinical outcomes, and rate of pharmacist intervention. This was a multi-center, retrospective study of adult patients admitted with MSSA or SOSA-positive blood cultures. The primary endpoint, days of vancomycin therapy, was compared between the pre- and post-BCID2 periods. Secondary endpoints included duration of MSSA bacteremia, hospital mortality, and pharmacist interventions. Of the 617 patients included in the primary analysis, MSSA bloodstream infection (BSI) accounted for 37.0% (110/297) and 31.6% (101/320) of the pre- and post-BCID2 group, respectively ($P = 0.178$). The median duration of vancomycin was 1.3 and 0.3 days in the pre- and post-BCID2 groups, respectively (absolute difference, 1.0 days; $P < 0.001$). The duration of MSSA BSI was 3.1 and 2.4 days in the pre- and post-BCID2 group, respectively (absolute difference, 0.7 days; $P = 0.096$). Hospital mortality was not different between periods (7.1% [21/297] vs 10.6% [34/320], $P = 0.16$). Pharmacists successfully intervened to reduce vancomycin duration in 7.1% (21/297) and 16.3% (52/320) of patients in the pre- and post-BCID2 group, respectively ($P < 0.001$). BCID2 implementation shortened the duration of vancomycin in patients with MSSA or SOSA bacteremia. These data support BCID2 as a critical tool to improve appropriate antibiotic selection and shorten the time to safer and more effective treatment for patients with SOSA and MSSA bacteremia.

**IMPORTANCE** PCR-based blood culture testing with concurrent antimicrobial stewardship program review continues to facilitate the initiation of pathogen-targeted antimicrobials in a shorter amount of time compared to traditional methods. The initiation of pathogen-targeted antimicrobials compared to non-pathogen-targeted antimicrobials has been shown to improve clinical outcomes in bloodstream infections caused by specific pathogens (e.g., methicillin-susceptible *Staphylococcus aureus* [MSSA]). Additionally, the adverse effects of vancomycin may be decreased with a shorter duration in blood culture contamination caused by coagulase-negative *Staphylococcus* species. This retrospective study evaluated the amount of time BCID2 results with pharmacist review and intervention reduced time to pathogen-targeted antimicrobials in MSSA, and time to discontinuation in blood culture contamination by coagulase-negative *Staphylococcus* species. Secondary outcomes mainly evaluated the clinical effects of these interventions.

**KEYWORDS**   bacteremia, rapid diagnostic tests, bloodstream infections, antimicrobial stewardship

Antimicrobial stewardship programs (ASPs) rely heavily on collaboration with microbiology laboratory services to effectively implement initiatives to improve

Address correspondence to Colin H. Duell, duellc1@gmail.com.

The authors declare no conflict of interest.

antimicrobial use. Bacteremia represents a potentially high-impact disease state warranting ASP review, as delays in appropriate antibiotic therapy are associated with higher mortality (1). Prompt pathogen identification and initiation of pathogen-directed therapies are often delayed due to traditional microbiologic techniques, which depend on bacterial growth on culture media. To ensure appropriate coverage initially, empiric antimicrobial regimens are often excessively broad, which may precipitate future antimicrobial resistance and contribute to antimicrobial overuse. For these reasons, rapid identification technologies have gained traction as ASP tools to appropriately escalate or de-escalate therapy based on the pathogen's genetic information.

BioFire Blood Culture Identification 2 (BCID2) panel is a multiplex polymerase chain reaction (PCR)-based technology designed to extract, amplify, and detect nucleic acid from a positive blood culture (2). This panel can identify 43 frequently encountered bacterial and yeast species, while also identifying key genetic determinants of resistance that may influence drug selection, including mecA and mecC. BCID2 allows for differentiation of methicillin-resistant *Staphylococcus aureus* (MRSA), methicillin-susceptible *S. aureus* (MSSA), and Staphylococci other than *S. aureus* (SOSA). BCID2 will result as MRSA only if all of the following targets are detected: MREJ, mecA/C, and *S. aureus*. BCID2 will result as MSSA only if *S. aureus* is detected in the absence of mecA/C. The only SOSA organisms that will be called mecA-positive are *Staphylococcus epidermidis* and *Staphylococcus lugdunensis*. Results are available faster than traditional techniques, generally an hour after bacterial growth is detected in a blood culture bottle.

Given that BCID2 can readily identify MSSA, antimicrobial therapy can be tailored to the most appropriate treatment option, namely, cefazolin or an anti-staphylococcal penicillin. Transition to MSSA-targeted agents has been shown to reduce mortality when compared to vancomycin alone (3). Blood culture contamination may confuse the clinical picture surrounding bacteremia, potentially leading to unnecessary antibiotic therapy and extending hospital length of stay (LOS). SOSA represent one group of potential bacterial contaminants that can be identified by BCID2 and account for 75%–88% of contaminated blood cultures (4).

With an ongoing need for enhanced stewardship strategies, utilization of rapid diagnostics can improve antimicrobial prescribing while improving clinical outcomes when added to an ASP's workflow. In September 2022, Hartford HealthCare added BCID2 as part of the standard work for positive blood cultures. Once blood culture PCR results become available, they are batched and sent via e-mail to ASP pharmacists in 3 hour intervals (i.e., 0600, 0900, 1200, 1500, 2100), allowing formal review during usual work hours. This study evaluated the duration of vancomycin, clinical outcomes, and rate of pharmacist intervention in hospitalized patients with MSSA or SOSA-positive blood cultures, before and after implementation.

## MATERIALS AND METHODS

This is a multi-center, quasi-experimental, retrospective study of records of adult patients admitted to any of the seven hospitals within a large healthcare system with blood cultures growing either MSSA or a SOSA species, with the exception of *S. lugdunensis*. *S. lugdunensis* was excluded due to its high virulence compared to other SOSA. The pre-implementation period was defined as the 6 months prior to BCID2 roll-out (March 7, 2022–September 7, 2022) and the post-implementation period was defined as 6 months, starting 1 month after the roll-out date and continuing through 7 months after the roll-out date (October 7, 2022–April 7, 2023).

Patients were excluded from the analysis if they had polymicrobial bacteremia, had blood cultures collected or antimicrobial treatment started prior to hospitalization at a HHC hospital, survived <24 hours after BCID2 results became available, were transitioned to comfort measures only within the first 48 hours of bacteremia treatment, were treated with an anti-MSSA antibiotic prior to BCID2 results (cefazolin, nafcillin, or ampicillin/sulbactam), had SOSA-positive blood cultures and received vancomycin for a concomitant

infection, or were not started on treatment within 24 hours of blood cultures being taken for patients with eventual MSSA bacteremia.

Using the electronic medical record, patients were identified via Epic reports based on BCID2 and blood culture and susceptibility results that identified MSSA and SOSA within the defined study timeframe. Reports included patient demographics, hospital admission and LOS data, date of culture positivity, Charlson comorbidity index, and date of death where applicable. Additional electronic chart review was conducted to collect information on past medical history including risk factors for bacteremia (current or former person who injects drugs (PWID), history of bloodstream infection (BSI) caused by a gram-positive bacterium, presence of prosthetic heart valves or implantable cardiac device(s), current chronic hemodialysis patient, and IV antibiotic use in the last 90 days, antibiotic allergies, timing of blood culture draws and results, antibiotic treatment, pharmacist interventions, vasopressor utilization, readmission data, serum creatinine (SCr) trends, and rate of *Clostridioides difficile* infection (CDI) at 30 days (5, 6).

The primary outcome, duration of empiric vancomycin, was compared between patients in the pre- and post-BCID2 implementation periods. Vancomycin duration was measured from the administration of the first dose of vancomycin to the discontinuation of either the vancomycin maintenance regimen order or the pharmacy to dose vancomycin protocol, whichever was continued for longer. Every patient who received vancomycin had a "pharmacy to dose" protocol ordered concurrently. Patients who did not receive any IV vancomycin were counted as 0 days of antibiotic therapy. Subgroup analyses were conducted comparing patients with SOSA- and MSSA-positive blood cultures who received vancomycin.

Secondary outcomes consisted of hospital LOS, time to blood culture results, CDI and 30-day hospital readmission, accepted pharmacist interventions that shortened the duration of vancomycin, and incidence of acute kidney injury (AKI). Incidence of AKI was defined as an increase in SCr of at least 1.5 times their baseline while receiving vancomycin, consistent with risk, injury, failure, loss of kidney function, and end-stage kidney disease (RIFLE)'s "Risk" criteria. RIFLE's "Injury" criteria of an increase in SCr of at least two times the baseline SCr while receiving vancomycin was also assessed (7). Time to blood culture results were defined as the amount of time elapsed between blood culture collection and definitive results (i.e., culture and susceptibility reports in the pre-BCID2 group and PCR results in the post-BCID2 group). Accepted pharmacist interventions were made based on PCR results in the post-BCID2 group. In the MSSA cohort, additional outcomes were collected, including in-hospital mortality, time to MSSA-targeted agent, duration of bacteremia, ICU LOS, and duration of IV vasopressors.

A power and sample size analysis estimated that 800 MSSA and SOSA blood cultures would be distributed as 25%–30% and 70%–75% of each cohort during the two 6-month periods, respectively. Using a baseline duration of vancomycin therapy of 72 hours and a 12-hour difference (reduction) as a meaningful clinical difference, a sample of 100 MSSA and 300 SOSA blood cultures in each cohort would afford 80% power using a Student's *t*-test with an alpha of 0.05 and a common standard deviation (SD) of 30 and 52 hours, respectively. Continuous data were evaluated for normality of distribution. Continuous, normally distributed variables are presented as means and SDs and compared with a Student *t*-test. Non-normally distributed variables are presented as medians and interquartile ranges (IQRs) and compared with a Mann-Whitney *U* test. Categorical variables are presented as frequencies, using percentages, and evaluated with a Pearson $\chi^2$ test or Fisher's exact test where appropriate. To evaluate the impact of differences in baseline characteristics, analyses of covariance were conducted using a one-way analysis of covariance (ANCOVA) test. All analyses were conducted with SPSS v. 29 (IBM; Armonk, NY 2022) using an alpha level of 0.05 such that all results yielding $P < 0.05$ will be deemed statistically significant.

## RESULTS

A total of 1,285 patient charts were reviewed for eligibility, of which 617 were included in the primary analysis with 51.9% (320/617) and 48.1% (297/617) patients comprising the pre- and post-BCID2 group, respectively. The primary reason for exclusion besides duplicate patients (i.e., patients who met inclusion criteria more than once within the study period) was patients with SOSA-positive blood culture(s) that received vancomycin for a concomitant infection (Fig. 1). Baseline characteristics, including risk factors for gram-positive bacteremia, were similar between the two groups, with the exception of sex. There were significantly more females in the post-BCID2 group (54.4% [174/320] vs 43.1% [128/297], $P = 0.007$). The majority of patients who were Caucasian had a median age of 69, with a Charlson comorbidity index of 3 (IQR, 2–5). MSSA bacteremia accounted for 37.0% (110/297) and 31.6% (101/320) of the patients in the pre- and post-BCID2 group, respectively ($P = 0.178$) (Table 1).

The median duration of vancomycin was 1.3 and 0.3 days for all patients, including those who did not receive empiric vancomycin (absolute difference, 1.0 days; $P < 0.001$), and 2.1 and 1.0 days in patients who received empiric vancomycin in the pre- and post-BCID2 group, respectively (absolute difference, 1.1 days; $P < 0.001$). The median duration of vancomycin was 1.7 and 1.0 days in patients with SOSA-positive blood cultures who received vancomycin (absolute difference, 0.7 days; $P = 0.005$) and 2.2 and

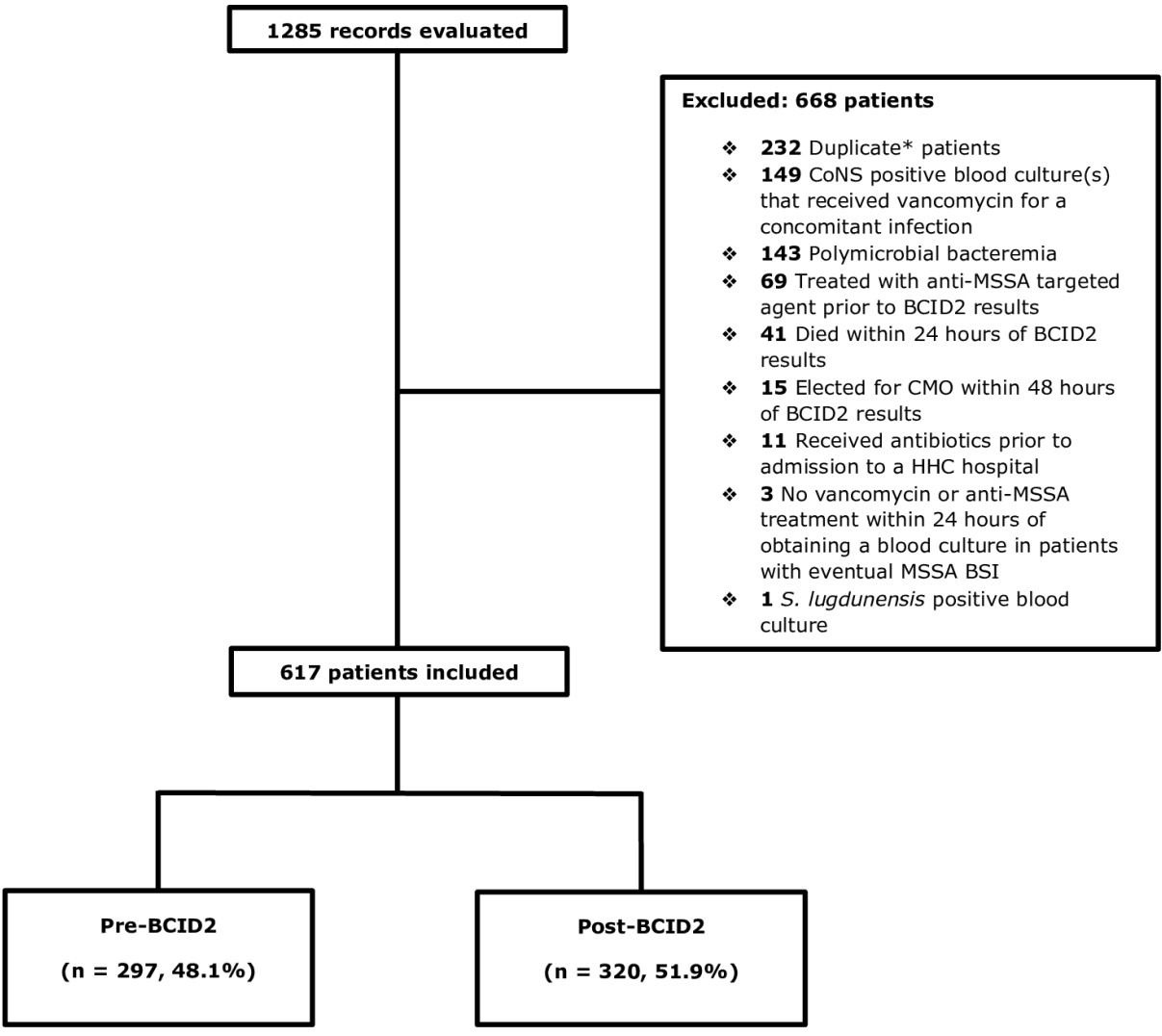

**FIG 1** CONSORT diagram for study. *Duplicate patients: patients who met inclusion criteria more than once within the study period.

**TABLE 1** Baseline characteristics[a]

| Baseline characteristics | Pre-BCID2 (n = 297) | Post-BCID2 (n = 320) | P value |
|---|---|---|---|
| Sex (no., %) | | | |
| Female | 128/297 (43.1%) | 174/320 (54.4%) | 0.007 |
| Age, years (median [IQR]) | 68 (57–80) | 70 (60–80) | 0.379 |
| Body weight, kg (median [IQR]) | 78.6 (64.1–96.7) | 79.5 (65.8–94.9) | 0.876 |
| BMI, kg/m$^2$ (median [IQR]) | 27.8 (23.1–33.4) | 28.7 (24.4–33.6) | 0.157 |
| Race/ethnicity (no., %) | | | 0.326 |
| Caucasian | 234 (78.8%) | 256 (81.5%) | |
| Hispanic | 35 (11.8%) | 34 (10.8%) | |
| Black | 24 (8.1%) | 24 (7.6%) | |
| Asian | 2 (0.7%) | 0 (0.0%) | |
| Pacific Islander | 1 (0.3%) | 0 (0.0%) | |
| American Indian or Alaskan | 1 (0.3%) | 0 (0.0%) | |
| Ethnicity (no., %) | | | |
| Hispanic or Latino | 35 (11.8%) | 34 (10.8%) | 0.806 |
| Location at time of blood culture (no., %) | | | 0.752 |
| ED | 251 (84.5%) | 265 (82.8%) | |
| Ward or stepdown unit | 29 (9.8%) | 32 (10.0%) | |
| ICU | 17 (5.7%) | 23 (7.2%) | |
| Beta-lactam allergy, n (%) | 53 (17.8%) | 54 (16.9%) | 0.95 |
| CCI (median [IQR]) | 3.0 (2–5) | 3.0 (2–5) | 0.713 |
| Required ICU admission during hospital stay (no., %) | | | |
| Yes | 35 (39.5%) | 47 (42.5%) | 0.289 |
| Blood culture collected in the ICU, n (%) | 17 (5.7%) | 23 (7.2%) | 0.752 |
| MSSA bacteremia, n (%) | 110 (37.0%) | 101 (31.6%) | 0.178 |
| Current or former PWID (no., %) | | | |
| Yes | 19 (6.4%) | 15 (4.7%) | 0.451 |
| History of gram-positive BSI (no., %) | | | |
| Yes | 30 (10.1%) | 35 (10.9%) | 0.836 |
| History of gram-positive IE (no., %) | | | |
| Yes | 5 (1.7%) | 4 (1.3%) | 0.91 |
| Presence of prosthetic heart valve or ICD (no., %) | | | |
| Yes | 45 (15.2%) | 34 (10.6%) | 0.119 |
| Receiving chronic HD (no., %) | | | |
| Yes | 15 (5.1%) | 16 (5.0%) | 0.876 |
| History of IV antibiotics in the past 90 days (no., %) | | | |
| Yes | 62 (20.9%) | 240 (75.0%) | 0.263 |

[a]BMI, body mass index; BSI, blood stream infection; CCI, Charlson comorbidity index; ED, emergency department; HD, hemodialysis; ICD, intra-cardiac device; ICU, intensive care unit; IE, infective endocarditis; IQR, interquartile range; IV, intravenous; MSSA, methicillin-susceptible *S. aureus*; PWID, person who injects drugs.

1.0 in patients with MSSA bacteremia who received vancomycin (absolute difference, 1.2 days; $P < 0.001$). The number of patients who received at least one dose of vancomycin was 216 (72.7%) and 201 (62.8%) in the pre- and post-BCID2 group, respectively. Accepted pharmacist interventions that shortened the duration of vancomycin occurred in 7.1% (21/297) and 22.5% (73/320) in the pre- and post-BCID2 group, respectively ($P < 0.001$). Time to blood culture results was 2.5 and 1.2 days in the pre- and post-BCID2 group, respectively (absolute difference, 1.3 days; $P < 0.001$). Hospital LOS (6.8 vs 6.7 days, $P = 0.766$), hospital readmission at 30 days (16.6% vs 15.6%, $P = 0.838$), and incidence of CDI at 30 days (1.5% vs 1.0%, $P = 0.958$) were not statistically different between the two groups. Incidence of AKI was lower in the pre-BCID2 group, with 5.0% (10/200) incurring an AKI, compared to 12.3% (23/187) in the post-BCID2 group ($P = 0.017$). Given this finding, a post-priori analysis was conducted in patients who developed an AKI. In this AKI cohort, the duration of vancomycin was longer in the post-BCID2 group (2.0 vs 1.5 days, $P = 0.003$). Additionally, when the definition of AKI was adjusted to the

**TABLE 2** Overall results[a]

| Result | Pre-BCID2 (n = 297) | Post-BCID2 (n = 320) | P value |
|---|---|---|---|
| Duration of vancomycin, days (median [IQR]) | 1.3 (0.0–2.6) | 0.3 (0.0–1.5) | <0.001 |
| Duration of vancomycin in patients who received vancomycin, days (median [IQR]) | 2.1 (1.1–3.0) | 1.0 (0.5–1.9) | <0.001 |
| Duration of vancomycin in patients with CoNS positive blood culture(s) who received vancomycin, days (median [IQR]) | 1.7 (0.8–3.2) | 1.0 (0.3–2.1) | 0.005 |
| Duration of vancomycin in patients with MSSA BSI who received vancomycin, days (median [IQR]) | 2.2 (1.6–3.0) | 1.0 (0.7–1.7) | <0.001 |
| Hospital LOS, days (median [IQR]) | 6.8 (3.9–11.6) | 6.7 (3.6–11.1) | 0.766 |
| Incidence of AKI using RIFLE "Risk" definition (no., %) | 10 (5.0%) | 23 (12.3%) | 0.017 |
| Incidence of AKI using RIFLE "Injury" definition (no., %) | 6 (3.0%) | 9 (4.8%) | 0.356 |
| Time to blood culture result, days (median [IQR]) | 2.5 (2.0–2.9) | 1.2 (1.0–1.6) | <0.001 |
| Accepted pharmacist interventions that shortened duration of vancomycin, (no., %) | 21 (7.1%) | 73 (22.5%) | <0.001 |
| CDI at 30 days, (no., %) | 4 (1.5%) | 3 (1.0%) | 0.958 |
| Hospital readmission at 30 days, days (median [IQR]) | 45 (16.6%) | 44 (15.6%) | 0.838 |

[a]AKI, acute kidney injury; BSI, blood stream infection; CDI, *Clostridium difficile* infection; IQR, interquartile range; LOS, length of stay; MSSA, methicillin-susceptible *S. aureus*; RIFLE, Risk, Injury, Failure, Loss of kidney function, and End-stage kidney disease.

RIFLE "Injury" criteria of an increase in baseline SCr by two times, there was no statistical difference in AKI between the groups (3.0% [6/200] vs 4.8% [9/187], P = 0.356) (Table 2).

In patients with MSSA BSI, the duration of MSSA BSI was 3.1 and 2.4 days in the pre- and post-BCID2 group, respectively (absolute difference, 0.7 days; P = 0.096). Time to MSSA-targeted agent was 2.9 and 1.6 days in the pre- and post-BCID2 group, respectively (absolute difference, 1.3 days; P < 0.001). ICU LOS was 4.4 days shorter in the post-BCID2 group (7.8 vs 3.4 days, P < 0.001), and in-hospital mortality was not different between groups (7.1% [21/297] vs 10.6% [34/320], P = 0.16) (Table 3).

Given the statistically significant difference in sex between the two study groups, the impacts of sex on all primary, secondary, and additional secondary endpoints were evaluated post-priori using female vs male sex as populations. Statistical differences between the two sexes were observed in two outcomes: overall vancomycin duration and hospital LOS. An ANCOVA was conducted using a one-way ANCOVA test and found that the difference seen in the pre- vs post-BCID2 groups was independent of sex. (P = 0.301 for vancomycin duration, P = 0.475 for hospital LOS).

## DISCUSSION

In this retrospective analysis, BCID2 implementation led to a significant decrease in the duration of vancomycin in patients with MSSA- or SOSA-positive blood cultures. This finding was supported by subgroup analyses that included only patients who received empiric vancomycin, as well as patients with MSSA- and SOSA-positive blood cultures who received vancomycin. Interventions to limit vancomycin use were pharmacist intervention and/or ID consultation in both the pre- and post-BCID2 groups. To our

**TABLE 3** MSSA-specific results[a]

| Result | Pre-BCID2 (n = 110) | Post-BCID2 (n = 101) | P value |
|---|---|---|---|
| In-hospital mortality (no., %) | 9 (8.2%) | 13 (12.9%) | 0.375 |
| Time to MSSA-targeted agent, days (median [IQR]) | 2.9 (2.2–3.3) | 1.6 (1.0–2.2) | <0.001 |
| Duration of bacteremia, days (median [IQR]) | 3.1 (1.8–4.4) | 2.4 (1.6–4.1) | 0.091 |
| ICU LOS, days (median [IQR]) | 7.8 (3.4–29) | 3.4 (2.0–6.3) | <0.001 |
| Duration of IV vasopressors, days (median [IQR]) | 2.2 (1.0–8.4) | 1.4 (0.4–3.3) | 0.061 |

[a]ICU, intensive care unit; IQR, interquartile range; IV, intravenous; LOS, length of stay; MSSA, methicillin-susceptible *S. aureus*.

knowledge, this is the first study to evaluate the effect of PCR-based testing in patients with MSSA and SOSA bacteremia without including MRSA bacteremia.

Regarding the duration of vancomycin and time to MSSA-targeted agent, our antimicrobial use data are in line with previous data. MacVane et al. assessed the effects of a PCR-based blood culture identification panel (BCID) on time to effective therapy and rates of antimicrobial de-escalation in any bacteremia caused by a pathogen that was able to be detected by BCID, of which 16.2% (59/364) of the population had a MSSA- or SOSA-positive blood culture. Utilization of BCID led to a significant decrease of 10 hours when evaluating time to effective therapy and 15 hours in time to de-escalation when compared to patients that did not have BCID results (8). More specific to Staphylococcal bacteremia, Baur et al. (9) assessed similar outcomes after PCR implementation and found a shorter mean time to MSSA-targeted therapy by 1.7 days after PCR implementation. More recently, Goshorn et al. evaluated the effects of PCR-based blood culture testing in tandem with an ASP-driven management algorithm for SOSA bacteremia. They found that empiric vancomycin therapy was avoided and shortened by a median of 1.2 days in the post-PCR cohort (10). Importantly, the intervention of PCR testing was augmented by an ASP in these three studies, each finding significantly shorter time to optimal therapy in Staphylococcal bacteremia.

Time to optimal treatment after implementation of PCR-based blood culture testing has also been evaluated independent of ASPs. Emonet et al., Bukowski et al., and Frye et al. evaluated similar endpoints with PCR-based blood culture identification in adult inpatients who had blood cultures positive for gram-positive cocci (GPC) in clusters. Emonet et al. (11) identified a shorter time to targeted treatment in MSSA and MRSA, but not when all *Staphylococcus* species were included. In Bukowski et al. (12), time to appropriate therapy was significantly shortened in the total cohort; however, time to MRSA- or MSSA-preferred therapy was not. Frye et al. (13) found that the time to optimal therapy was not significantly reduced in any bacteremia caused by GPC in clusters. These findings are conflicting and contradict findings from studies that include PCR implantation alongside antimicrobial stewardship efforts. This highlights the importance of ASPs and the role of the pharmacist to attain beneficial antimicrobial use outcomes. In our study, BCID2 implementation was found to facilitate pharmacist intervention to switch or stop empiric vancomycin, suggesting that pharmacists recommended switching vancomycin to a MSSA-targeted agent in MSSA BSI or discontinuing vancomycin in SOSA-positive blood cultures that were deemed contaminant. The difference seen in accepted pharmacist interventions that shortened the duration of vancomycin was most likely due to less opportunity for pharmacist intervention in the pre-BCID2 period, as providers were used to changing or discontinuing vancomycin when matrix-assisted laser desorption ionisation-time of flight(MALDI-TOF) results were available. When BCID2 was implemented in our health system, this was a novel test to interpret and respond to.

Microbiological outcomes were also evaluated in some of the aforementioned studies. The significant reduction in time to blood culture results of 1.3 days found in our evaluation was similar to the 0.6-day reduction from time to organism ID, 0.9-day reduction in time from gram stain to methicillin susceptibility results, and 1.7-day reduction in time to identification found in MacVane et al., Emonet et al., and Frye et al., respectively. The discordance between the findings of MacVane et al. and Frye et al. may be due to the availability of MALDI, with hospitals that do not have access to MALDI likely experiencing a greater benefit with BCID2 results (8, 11, 13).

Previous studies have evaluated the effect of PCR-based blood culture testing on mortality, 30-day readmission, ICU LOS, hospital LOS, relapse of BSI within 90 days, hospital onset CDI, and vancomycin-induced AKI in patients with Staphylococcal bacteremia. Bukowski et al. (12) is the only study identified as having found a statistically significant difference in any clinical outcome, finding significant reduction in hospital LOS in patients with SOSA-positive blood cultures that were deemed contaminant when BCID was utilized. When contrasted with the findings of our study, we found a significant difference in ICU LOS in patients with MSSA bacteremia, although our study

included BCID2 implementation alongside antimicrobial stewardship efforts. A clinically significant finding of this study is the effect of BCID2 on duration of MSSA bacteremia. Although not statistically significant, our finding suggests that BCID2 implementation, in combination with antimicrobial stewardship efforts, may shorten the median duration of bacteremia by approximately 16 hours. In our study, BCID2 implementation did not show effect on other clinical outcomes including hospital LOS, hospital readmission at 30 days, CDI at 30 days, and duration of vasopressors in patients with MSSA bacteremia.

Despite the statistical reduction in vancomycin therapy, the incidence of AKI was higher in the post-BCID2, contradicting our original hypothesis. However, in a *post hoc* analysis, the duration of vancomycin was significantly longer in patients who had an increase in SCr of at least 1.5 times their baseline in the post- vs pre-BCID2 group, and there was no difference noted between the groups in patients who developed a SCr of at least two times their baseline. While this may have impacted incidence of vancomycin-associated AKI, elevations in SCr typically occur at least 4 days after initiating vancomycin, and patients who incurred an AKI in the post-BCID2 groups only received vancomycin for a median of 2 days (14). Higher rates of AKI may also be explained by severity of illness, as the post-BCID2 group had numerically more patients having blood cultures collected in the ICU, and numerically more patients requiring ICU admission at some point during their stay, suggesting a sicker population more prone to developing an AKI. Additionally, the administration of concomitant nephrotoxic medications was not collected in our review.

Although no statistically significant difference was found in regard to in-hospital mortality, it was numerically higher in the post-BCID2 group. Further investigation was critical. The numerical increase may be driven by the difference in seasonality, given that the pre-BCID2 group contained admitted patients in the spring and summer months, while the post-BCID2 group contained admitted patients in the fall and winter months (15). Records were reviewed and found that in patients who died at 30 days, 13.6% (3/22) and 22.7% (10/44) in the pre- and post-BCID2 group had positive SARS-CoV2 nasal PCR swabs between the time they had positive blood cultures and the time that they died, respectively. With numerically more patients in the post-BCID2 group having an initial blood culture drawn in the ICU, this could suggest that the post-BCID2 group had more hospital-acquired infections compared to the pre-BCID2 group.

Limitations include the retrospective nature of this study, for which there are inherent biases. Certain variables may not have been recorded in our electronic medical record (EMR) and may have existed in the EMRs of other institutions without our knowledge (e.g., history of gram-positive BSI or IE). In the pre-BCID2 group, providers would commonly start dual MSSA/MRSA antibiotic therapy prior to culture and susceptibility result, which led to exclusion of those patients, and therefore a lower *n* value for the pre-BCID2 group. Regarding stewardship interventions, results were released to ASP pharmacists only during normal working hours, and only Monday-Friday at some sites within the health system. Additionally, pharmacists are not the only clinicians that conduct antimicrobial stewardship, and we did not capture interventions made by non-pharmacists. Lastly, our microbiology lab does not report the identification of mecA or mecC for SOSA, which may have increased the duration of vancomycin in the post-BCID2 group.

## Conclusions

BCID2 implementation shortened the duration of vancomycin in patients with MSSA or SOSA bacteremia and facilitated pharmacist intervention to stop vancomycin in non-pathogenic SOSA-positive blood cultures or switch to MSSA-targeted therapy in MSSA BSI. These data support BCID2 as a critical tool to improve appropriate antibiotic selection and shorten the time to safer and more effective treatment for patients with SOSA and MSSA bacteremia. The findings of this study will be used within our healthcare system to educate staff on how to interpret and act upon BCID2 results and facilitate BCID2 results outside of weekdays to involve clinical pharmacy staff to further elicit the

benefits found. Future studies evaluating the impact of pharmacist and ASP intervention would be beneficial to further evaluate effectiveness and guide allocation of ASP efforts.

## AUTHOR AFFILIATIONS

[1]Department of Pharmacy Services, Hartford Hospital, Hartford, Connecticut, USA

[2]Center for Anti-Infective Research and Development, Hartford Hospital, Hartford, Connecticut, USA

[3]School of Pharmacy and Physician Assistant Studies, The University of Saint Joseph, West Hartford, Connecticut, USA

[4]Department of Pharmacy Services, Hartford Healthcare, Hartford, Connecticut, USA

## AUTHOR ORCIDs

Colin H. Duell  http://orcid.org/0009-0000-9727-1242
Tyler Ackley  http://orcid.org/0000-0001-7977-5068
Joseph L. Kuti  http://orcid.org/0000-0002-4464-3126

## AUTHOR CONTRIBUTIONS

Colin H. Duell, Conceptualization, Data curation, Formal analysis, Investigation, Writing – original draft | Tyler Ackley, Conceptualization, Investigation, Project administration, Supervision, Writing – review and editing | Kristin E. Linder, Conceptualization, Project administration, Supervision, Writing – review and editing | Joseph L. Kuti, Conceptualization, Formal analysis, Project administration, Supervision, Writing – review and editing | Alivia Castle, Data curation, Investigation | Rosanna Li, Conceptualization, Formal analysis, Project administration, Supervision, Writing – review and editing

## ADDITIONAL FILES

The following material is available online.

### Open Peer Review

**PEER REVIEW HISTORY (review-history.pdf).** An accounting of the reviewer comments and feedback.

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
