## [Reviewer comments · Microbiology Spectrum]

Microbiology Spectrum

Clinical Impact of BioFire Blood Culture Identification 2 Implementation in Patients with Staphylococcus Bacteremia

Colin Duell, Tyler Ackley, Kristin Linder, Joseph Kuti, Alivia Castle, and Rosanna Li

Corresponding Author(s): Colin Duell, Hartford Hospital

Review Timeline:

Submission Date:	April 8, 2025
Editorial Decision:	June 10, 2025
Revision Received:	August 6, 2025
Editorial Decision:	August 11, 2025
Revision Received:	September 21, 2025
Accepted:	September 25, 2025

Editor: Daniel Ortiz

Reviewer(s): The reviewers have opted to remain anonymous.

Transaction Report:

DOI: <https://doi.org/10.1128/spectrum.01093-25>

Re: Spectrum01093-25 (Clinical Impact of BioFire Blood Culture Identification 2 Implementation in Patients with Staphylococcus Bacteremia)

Dear Dr. Colin Duell:

Thank you for the privilege of reviewing your work. Below you will find instructions from the Spectrum editorial office and the reviewer comments.

Revision Guidelines

Sincerely,
Daniel Ortiz
Editor
Microbiology Spectrum

Reviewer #1 (Comments for the Author):

This manuscript describes the comparison of antimicrobial therapy for MSSA and staphylococci other than aureus before and after the introduction of molecular testing that includes the identification of these organisms and stewardship intervention based on molecular results.

My comments are shown below.

1. Recommend updating CoNS terminology to CLSI recommendation of SOSA, staphylococci other than Staphylococcus aureus.

2. Please clarify the use of the MREJ target to differentiate mecA/C from SOSA vs. *S. aureus*.
3. Please provide details on how SOSA patients were clinically categorized as contamination vs. infection and how the option so of a mecA/C result for some SOSA was factored into therapy decisions in those determined to have infection rather than contamination. If this isn't being considered inclusion of SOSA in the manuscript is not meaningful.
4. Line 85 is confusing and may be missing words.

Reviewer #2 (Comments for the Author):

This is a well written and thoughtful study. Rapid ID and susceptibility has been touted as way to decrease time to appropriate antibiotics and thereby improve clinical outcomes. The authors are to be commended for contributing to the evidence base on this topic. I have a few questions for the authors.

1. Since clinical outcomes were not affected, did the 1 day decrease in Vanco doses given justify the use of an expensive Rapid ID/Susceptibility system? It would be good to consult the laboratory personnel and include their viewpoints on cost of the rapid ID/Susceptibility system and address the question- if there is no clinical effect is the cost justified? healthcare costs are increasing at a rapid pace, and healthcare is bankrupting patients and putting some hospitals out of business. I am not convinced that the data you have presented justifies the conclusion that you have stated in the paper.
2. More detail is needed on how many cases were called by the pharmacists in total versus not called and why and how many times did pharmacists intervene unsuccessfully which may help to address how well did the intervention work? and what could be done, if anything to improve the positive intervention rate?
3. Line 283, the word illicit should be elicit

Reviewer #1 (Comments for the Author):

1. Recommend updating CoNS terminology to CLSI recommendation of SOSA, staphylococci other than *Staphylococcus aureus*.

Done

2. Please clarify the use of the MREJ target to differentiate *mecA/C* from SOSA vs. *S. aureus*.

This was not discussed in our project. To my knowledge (please correct me if I am wrong), BCID2 detects genetic material of the organism itself (e.g., *Staphylococcus epidermidis* vs. *Staphylococcus aureus*), and does not rely solely on detection of MREJ to differentiate MR SOSA vs. MRSA.

Additionally, similar to our answer in question 3, when SOSA is detected by BCID2, the *mecA/C* gene identification result is hidden unless an ID pharmacist or ID provider requests that it is unhidden. This was done due to the concern of misinterpretation of *mecA/C* results and the concern for over treatment by incorrectly interpreting MRSA.

We can add the above statement into our methods section if you deem appropriate.

3. Please provide details on how SOSA patients were clinically categorized as contamination vs. infection and how the option so of a *mecA/C* result for some SOSA was factored into therapy decisions in those determined to have infection rather than contamination. If this isn't being considered inclusion of SOSA in the manuscript is not meaningful.

Determination of SOSA contamination vs pathogenicity was ultimately decided by the treating provider.

Similar to our answer in question 3, when SOSA is detected by BCID2, the *mecA/C* gene identification result is hidden unless an ID pharmacist or ID provider requests that it is unhidden. This was done due to the concern of misinterpretation of *mecA/C* results and the concern for over treatment by incorrectly interpreting MRSA.

I believe you are getting at whether or not we included patients who had SOSA positive blood cultures who were transitioned from vanco to cefazolin or an anti-Staphylococcal penicillin if *mecA/C* was not detected/shown. Good point made. We did not encounter this occurrence in our review, but will add that to our exclusion criteria to make it clear. Patients either continued vanco for possible true infection (very rarely) or vanco was discontinued upon learning that BCID2 detected SOSA (that was not *S. lugdunensis*).

We did not exclude patients who had true, pathogenic SOSA bacteremia in either the pre- or post-BCID2 group (although there were likely a very small amount). The idea was to include SOSA bacteremia, whether pathogenic or not, in both pre- and post-BCID2 groups because of the clinical uncertainty that comes with determining whether SOSA being present in the blood represents a skin contaminant or a pathogen.

We did collect the number of positive blood culture bottles and whether or not they were in the same set. We could report the number of bottles/same or different sets for the pre- and post-BCID2 group to use as a surrogate for likely contamination.

I do understand that we cannot say with certainty that the pre- and post-BCID2 groups are balanced in terms of quantity of patients with contaminant SOSA vs. pathogenic SOSA. We can include this in the limitations section if you deem appropriate.

4. Line 85 is confusing and may be missing words.

Fixed

Reviewer #2 (Comments for the Author):

1. Since clinical outcomes were not affected, did the 1 day decrease in Vanco doses given justify the use of an expensive Rapid ID/Susceptibility system? It would be good to consult the laboratory personnel and include their viewpoints on cost of the rapid ID/Susceptibility system and address the question- if there is no clinical effect is the cost justified? healthcare costs are increasing at a rapid pace, and healthcare is bankrupting patients and putting some hospitals out of business. I am not convinced that the data you have presented justifies the conclusion that you have stated in the paper.

Fair point, I think the best justification in our analysis can be seen in the MSSA-specific results. Unfortunately, the In-hospital mortality outcome is confounded by what we believe to be seasonality. However, looking at the reduction in ICU LOS, and an ICU day ranging from \$2,000-10,000 per day, would say there is likely cost savings there. The duration of bacteremia also trended towards fewer days in the post-BCID2 cohort.

The time to MSSA-targeted agents is significantly shorter. When paired with others that show mortality benefit (McDaniel, 2016) in MSSA BSI when cefazolin/anti-Staph penicillin is compared to vanco. Definitely can't say that the reduction in time means less mortality in our study though.

2. More detail is needed on how many cases were called by the pharmacists in total versus not called and why and how many times did pharmacists intervene

unsuccessfully which may help to address how well did the intervention work? and what could be done, if anything to improve the positive intervention rate?

We did collect the total number of pharmacist interventions and the % accepted. We can report/mention that in the results section. The reasons why pharmacists would not make an intervention was if vancomycin had already been switched (MSSA) or stopped (most SOSA) or if the pharmacist was not there/had the time/resources to review (overnights, evenings, weekends, and/or holidays).

I will say based on a first look, the pharmacist intervention acceptance rate is decent in both groups. 21/24 (87.5%) in the pre-BCID2 group and 73/78 (93.6%) in the post-BCID2 group.

We did not collect data on why the intervention was not accepted. I could hypothesize that some providers struggled with patients with penicillin allergies ordering cefazolin, as not every provider was familiar with the idea that the R1 side chain mediates cross reactivity and cefazolin has a structurally unique R1 side chain compared to other beta-lactams including penicillins. Other than that, I could also see providers being hesitant to discontinue vancomycin in patients with SOSA positive blood cultures.

It is important to note that we counted the intervention as “accepted” if the proposed changes were made within 24 hours of the intervention.

We can include the above in the results/discussion section if you deem appropriate.

3. Line 283, the word illicit should be elicit

Fixed

Re: Spectrum01093-25R1 (Clinical Impact of BioFire Blood Culture Identification 2 Implementation in Patients with Staphylococcus Bacteremia)

Dear Dr. Colin Duell:

Thank you for the privilege of reviewing your work. Below you will find my comments, instructions from the Spectrum editorial office, and the reviewer comments.

The reviewer comments were made to provide clarity and address pitfalls. Please make suggested changes provided by the reviewers in the manuscript itself with track changes.

To clarify Reviewer #1 Question #2: BCID2 releases an MRSA result only if all of the following targets are detected: MREJ, mecA/C, and *S. aureus* (see package insert info below). The only SOSA organisms that will be called mecA positive by the BioFire are *S. epidermidis* and *S. lugdunensis*. Please incorporate this explanation into the manuscript to avoid confusion among readers.

From the BCID2 package insert: mecA/C and MREJ (MRSA) - The SCCmec cassette integrates into a specific region in the Staphylococcus genome^{97,98}. In *S. aureus*, this insertion creates MREJ (SCCmec right-extremity junction), and molecular identification of this junction region provides specific identification of an *S. aureus* that carries the SCCmec cassette. A combined molecular detection of mecA/C, MREJ, and *S. aureus* indicates MRSA. However, it is possible for *S. aureus* to carry SCCmec that has lost the mecA/C gene (an 'empty cassette', estimated to be 3.9-5% of methicillin-susceptible *S. aureus*^{99,100}); such a strain would be a methicillin-susceptible *S. aureus* but could be misidentified by molecular methods if there is a co-detection of an additional Staphylococcus spp. that carries the mecA/C gene. The junction, or point of insertion of the SCCmec cassette, can lead to a variety of MREJ types (i-xxi).

Additional comments to address:

Line 84: There are 43 total targets on the panel which include organisms and resistance markers.

Lines 87-88: See above. MSSA detection by BCID2 is the absence of mecA/C and MREJ.

Table 1: Please include MRSA and SOSA bacteremia numbers as well.

Line 167: Unclear why there are two different medians of vancomycin for all patients (1.3 and 0.3 vs. 2.1 and 1.0). Does the second set of medians (2.1 and 1.0) represent patients not empirically started on vancomycin? Please clarify.

Lines 169-170: For SOSA positive blood cultures, what interventions were used to limit vancomycin usage? Physician education, pharmacy intervention, interpretive comments, etc. This needs to be expanded upon in the manuscript.

Lines 172-173: Was the reduction in overall vancomycin usage primarily attributed to de-escalation in MSSA patients?

Lines 173-174: Why was pharmacist intervention less successful when culture results were available during the pre-BCID2 period?

Line 175: In the methods section, the authors state "Time to blood culture results was defined as the amount of time elapsed between blood culture collection and definitive results (i.e., culture and susceptibility reports in the pre-BCID2 group and PCR results in the post-BCID2 group)." Since the implementation of BCID2 should not directly affect the timing of culture and susceptibility reports, were there other laboratory practice changes that contributed to culture results being available more quickly?

Lines 185-189: This is a great breakdown of the MSSA cohort. Why was a similar breakdown not conducted for SOSA patients?

Revision Guidelines

- Upload point-by-point responses to the issues raised by the reviewers in a file named "Response to Reviewers," NOT in your

cover letter.

- Upload a compare copy of the manuscript (without figures) as a "Marked-Up Manuscript" file.
- Upload a clean .DOC/.DOCX version of the revised manuscript and remove the previous version.
- Each figure must be uploaded as a separate, editable, high-resolution file (TIFF or EPS preferred), and any multipanel figures must be assembled into one file.
- Any supplemental material intended for posting by ASM should be uploaded with their legends separate from the main manuscript. You can combine all supplemental material into one file (preferred) or split it into a maximum of 10 files with all associated legends included.

Sincerely,
Daniel Ortiz
Editor
Microbiology Spectrum

Reviewer #1 (Comments for the Author):

1. Recommend updating CoNS terminology to CLSI recommendation of SOSA, staphylococci other than *Staphylococcus aureus*.

Done

2. Please clarify the use of the MREJ target to differentiate *mecA/C* from SOSA vs. *S. aureus*.

This was not discussed in our project. To my knowledge (please correct me if I am wrong), BCID2 detects genetic material of the organism itself (e.g., *Staphylococcus epidermidis* vs. *Staphylococcus aureus*), and does not rely solely on detection of MREJ to differentiate MR SOSA vs. MRSA.

Additionally, similar to our answer in question 3, when SOSA is detected by BCID2, the *mecA/C* gene identification result is hidden unless an ID pharmacist or ID provider requests that it is unhidden. This was done due to the concern of misinterpretation of *mecA/C* results and the concern for over treatment by incorrectly interpreting MRSA.

We can add the above statement into our methods section if you deem appropriate.

3. Please provide details on how SOSA patients were clinically categorized as contamination vs. infection and how the option so of a *mecA/C* result for some SOSA was factored into therapy decisions in those determined to have infection rather than contamination. If this isn't being considered inclusion of SOSA in the manuscript is not meaningful.

Determination of SOSA contamination vs pathogenicity was ultimately decided by the treating provider.

Similar to our answer in question 3, when SOSA is detected by BCID2, the *mecA/C* gene identification result is hidden unless an ID pharmacist or ID provider requests that it is unhidden. This was done due to the concern of misinterpretation of *mecA/C* results and the concern for over treatment by incorrectly interpreting MRSA.

I believe you are getting at whether or not we included patients who had SOSA positive blood cultures who were transitioned from vanco to cefazolin or an anti-Staphylococcal penicillin if *mecA/C* was not detected/shown. Good point made. We did not encounter this occurrence in our review, but will add that to our exclusion criteria to make it clear. Patients either continued vanco for possible true infection (very rarely) or vanco was discontinued upon learning that BCID2 detected SOSA (that was not *S. lugdunensis*).

We did not exclude patients who had true, pathogenic SOSA bacteremia in either the pre- or post-BCID2 group (although there were likely a very small amount). The idea was to include SOSA bacteremia, whether pathogenic or not, in both pre- and post-BCID2 groups because of the clinical uncertainty that comes with determining whether SOSA being present in the blood represents a skin contaminant or a pathogen.

We did collect the number of positive blood culture bottles and whether or not they were in the same set. We could report the number of bottles/same or different sets for the pre- and post-BCID2 group to use as a surrogate for likely contamination.

I do understand that we cannot say with certainty that the pre- and post-BCID2 groups are balanced in terms of quantity of patients with contaminant SOSA vs. pathogenic SOSA. We can include this in the limitations section if you deem appropriate.

4. Line 85 is confusing and may be missing words.

Fixed

Reviewer #2 (Comments for the Author):

1. Since clinical outcomes were not affected, did the 1 day decrease in Vanco doses given justify the use of an expensive Rapid ID/Susceptibility system? It would be good to consult the laboratory personnel and include their viewpoints on cost of the rapid ID/Susceptibility system and address the question- if there is no clinical effect is the cost justified? healthcare costs are increasing at a rapid pace, and healthcare is bankrupting patients and putting some hospitals out of business. I am not convinced that the data you have presented justifies the conclusion that you have stated in the paper.

Fair point, I think the best justification in our analysis can be seen in the MSSA-specific results. Unfortunately, the In-hospital mortality outcome is confounded by what we believe to be seasonality. However, looking at the reduction in ICU LOS, and an ICU day ranging from \$2,000-10,000 per day, would say there is likely cost savings there. The duration of bacteremia also trended towards fewer days in the post-BCID2 cohort.

The time to MSSA-targeted agents is significantly shorter. When paired with others that show mortality benefit (McDaniel, 2016) in MSSA BSI when cefazolin/anti-Staph penicillin is compared to vanco. Definitely can't say that the reduction in time means less mortality in our study though.

2. More detail is needed on how many cases were called by the pharmacists in total versus not called and why and how many times did pharmacists intervene

unsuccessfully which may help to address how well did the intervention work? and what could be done, if anything to improve the positive intervention rate?

We did collect the total number of pharmacist interventions and the % accepted. We can report/mention that in the results section. The reasons why pharmacists would not make an intervention was if vancomycin had already been switched (MSSA) or stopped (most SOSA) or if the pharmacist was not there/had the time/resources to review (overnights, evenings, weekends, and/or holidays).

I will say based on a first look, the pharmacist intervention acceptance rate is decent in both groups. 21/24 (87.5%) in the pre-BCID2 group and 73/78 (93.6%) in the post-BCID2 group.

We did not collect data on why the intervention was not accepted. I could hypothesize that some providers struggled with patients with penicillin allergies ordering cefazolin, as not every provider was familiar with the idea that the R1 side chain mediates cross reactivity and cefazolin has a structurally unique R1 side chain compared to other beta-lactams including penicillins. Other than that, I could also see providers being hesitant to discontinue vancomycin in patients with SOSA positive blood cultures.

It is important to note that we counted the intervention as “accepted” if the proposed changes were made within 24 hours of the intervention.

We can include the above in the results/discussion section if you deem appropriate.

3. Line 283, the word illicit should be elicit

Fixed

To clarify Reviewer #1 Question #2: BCID2 releases an MRSA result only if all of the following targets are detected: MREJ, mecA/C, and S. aureus (see package insert info below). The only SOSA organisms that will be called mecA positive by the BioFire are S. epidermidis and S. lugdunensis. Please incorporate this explanation into the manuscript to avoid confusion among readers.

Included, thank you for the information

Additional comments to address:

Line 84: There are 43 total targets on the panel which include organisms and resistance markers.

Included

Lines 87-88: See above. MSSA detection by BCID2 is the absence of mecA/C and MREJ.

Included, said "BCID2 will result as MSSA only if S. aureus is detected in the absence of mecA/C." because MREJ can be identified and still be MSSA (empty cassette).

Table 1: Please include MRSA and SOSA bacteremia numbers as well.

We did not gather the number of patients with MRSA bacteremia. Only MSSA and SOSA, making this an either/or scenario.

Line 167: Unclear why there are two different medians of vancomycin for all patients (1.3 and 0.3 vs. 2.1 and 1.0). Does the second set of medians (2.1 and 1.0) represent patients not empirically started on vancomycin? Please clarify.

Yes, exactly.

The first median is vancomycin duration in all patients, including patients who did not receive any vancomycin, with the thought being that perhaps because the BCID2 result was available before vancomycin could be started.

The second median is vancomycin duration in patients who actually received at least one dose of vancomycin, to show that even when vancomycin was started, the duration was still shorter in the post-BCID2 period.

Lines 169-170: For SOSA positive blood cultures, what interventions were used to limit vancomycin usage? Physician education, pharmacy intervention, interpretive comments, etc. This needs to be expanded upon in the manuscript.

Interventions to limit vancomycin use were pharmacist intervention and/or ID consultation in both the pre- and post-BCID2 groups. This is true for patients with SOSA or MSSA bacteremia. Included in the Discussion section.

Lines 172-173: Was the reduction in overall vancomycin usage primarily attributed to de-escalation in MSSA patients?

Good question, in our evaluation we did not determine which group (MSSA vs. SOSA) contributed to the overall decrease in vancomycin usage more. Hard to say with there being more SOSA patients in the study population, but a larger decrease in vancomycin usage in the MSSA group.

Lines 173-174: Why was pharmacist intervention less successful when culture results were available during the pre-BCID2 period?

Not sure that it was less successful, most likely just less opportunity for pharmacist intervention in the pre-BCID2 period, likely because providers were comfortable changing/discontinuing vancomycin when MALDI-TOF results were available and BCID2 being new to them. From the first review, the pharmacist intervention acceptance rate is decent in both groups. 21/24 (87.5%) in the pre-BCID2 group and 73/78 (93.6%) in the post-BCID2 group. Not a statistically significant difference.

Addressed in discussion section.

Line 175: In the methods section, the authors state "Time to blood culture results was defined as the amount of time elapsed between blood culture collection and definitive results (i.e., culture and susceptibility reports in the pre-BCID2 group and PCR results in the post-BCID2 group)." Since the implementation of BCID2 should not directly affect the timing of culture and susceptibility reports, were there other laboratory practice changes that contributed to culture results being available more quickly?

Sorry if this is worded in a confusing manner. Comparing time:

From blood culture collection to C+S report in pre-BCID2 group

From blood culture collection to BCID2 results in the post-BCID2 group.

The Time from blood culture collection to C+S report should remain the same between the two groups and was not collected nor assessed in the post-BCID2 group.

Lines 185-189: This is a great breakdown of the MSSA cohort. Why was a similar breakdown not conducted for SOSA patients?

Thank you. The same breakdown was not done in SOSA patients because we did not think that the MSSA endpoints were clinically relevant for SOSA patients, as we presumed the vast majority (if not all) of the SOSA patients represented blood culture contamination.

Re: Spectrum01093-25R2 (Clinical Impact of BioFire Blood Culture Identification 2 Implementation in Patients with Staphylococcus Bacteremia)

Dear Dr. Colin Duell:

Your manuscript has been accepted, and I am forwarding it to the ASM production staff for publication. Your paper will first be checked to make sure all elements meet the technical requirements. ASM staff will contact you if anything needs to be revised before copyediting and production can begin. Otherwise, you will be notified when your proofs are ready to be viewed.

Sincerely,
Daniel Ortiz
Editor
Microbiology Spectrum